# Fulvic Acid, Amino Acids, and Vermicompost Enhanced Yield and Improved Nutrient Profile of Soilless Iceberg Lettuce

**DOI:** 10.3390/plants14040609

**Published:** 2025-02-18

**Authors:** Beyza Keskin, Yelderem Akhoundnejad, Hayriye Yildiz Dasgan, Nazim S. Gruda

**Affiliations:** 1Department of Horticulture, Faculty of Agriculture, University of Cukurova, 01330 Adana, Türkiye; beyzaucel@gmail.com; 2Department of Horticulture, Faculty of Agriculture, University of Sirnak, 73000 Sirnak, Türkiye; yakhoundnejad@sirnak.edu.tr; 3Institute of Plant Sciences and Resource Conservation, Department of Horticultural Sciences, University of Bonn, D-53113 Bonn, Germany; ngruda@uni-bonn.de

**Keywords:** biostimulants, floating hydroponic culture, crop quality improvement, eco-friendly practices, *Lactuca sativa* var. *capitata*, nutritional quality, produce quality

## Abstract

Soilless cultivation systems are sustainable innovations in modern agriculture, promoting high efficiency per unit area, supporting food sustainability, and addressing the growing demand for high-quality produce with minimal environmental impact. This study evaluates the effects of fulvic acid, amino acid, and vermicompost biostimulants on the growth, yield, and nutrient profile of soilless-grown iceberg lettuce (*Lactuca sativa var*. *capitata)* in floating culture under controlled glasshouse conditions. Two experiments were conducted to determine the most effective concentrations and combinations of biostimulants. In the first experiment, varying doses of fulvic acid (40 and 80 ppm), amino acid (75 and 100 ppm), and vermicompost (1 and 2 mL L^−1^) were tested alongside a control. Optimal doses were identified based on their positive effects on lettuce growth and yield. The second experiment examined combinations of fulvic acid, amino acid, and vermicompost extract compared to a control. Biostimulants improved lettuce growth, nutrient uptake, and antioxidants. Vermicompost boosted root biomass and leaf area, while fulvic acid and amino acid reduced nitrates and increased dry matter. Fulvic acid and vermicompost resulted in the highest yield (17.15 kg/m^2^, 18.2% increase), and the combined treatment maximized antioxidants, increasing vitamin C by 17.16%, total phenols by 52.54%, and flavonoids by 52.38%. These findings highlight the potential of biostimulants as eco-friendly solutions for optimizing lettuce production in soilless systems.

## 1. Introduction

Soilless agriculture offers numerous advantages, including improved water-use efficiency, precise nutrient management, higher yields and quality, the ability to cultivate in regions with challenging conditions, reduced labor needs, enhanced irrigation practices, and streamlined production [1,2]. Soilless is a soil-free cultivation method extensively utilized for modern crop production on a global scale. This field encompasses various cultivation methods, such as deep-water culture, flowing water culture, nutrient film technique (NFT), and aeroponic culture [3,4]. In deep-water culture, plant roots are grown on a floating material like styrofoam, which is submerged in a nutrient-rich solution; this method is also known as floating hydroponic [5,6].

Lettuce (*Lactuca sativa* L.) belongs to the *Lactuca* genus within the Asteraceae family and is an annual vegetable. Its production occurs primarily in autumn, winter, and spring, as it is classified as a cool-season crop. Iceberg lettuce (*Lactuca sativa var. capitata*), commonly known as crisp iceberg lettuce, is a widely cultivated variety characterized by its compact, spherical heads and pale green, crisp leaves. It is unlike leaf lettuces (*Lactuca sativa var. crispa*), which have loose, ruffled leaves, or romaine lettuces (*Lactuca sativa var. longifolia*), known for their elongated, firm leaves. Iceberg lettuce is prized for its dense, crunchy texture and mild flavor [7]. These attributes make it particularly popular for adding texture to salads, sandwiches, and wraps. Iceberg lettuce is low in calories and rich in water. It is valued for its crisp texture and dietary fiber, vitamin A, vitamin K, folate (B9), vitamin C, potassium, and calcium [8]. It also contains antioxidants such as beta-carotene, lutein, and zeaxanthin [9]. Among lettuce types grown in soilless greenhouses and vertical indoor farming systems, iceberg lettuce stands out as particularly noteworthy.

Biostimulants are natural or synthetic substances that enhance plant growth, yield, and stress tolerance by stimulating physiological processes without acting as conventional nutrients or pesticides [10]. In agriculture, their primary purpose is to improve nutrient-use efficiency, enhance crop quality, and mitigate the effects of abiotic stresses, offering an innovative and sustainable approach to increasing productivity while reducing environmental impact [11,12,13].

Humic substances, among the most prominent biostimulants, are organic compounds naturally formed through the decomposition of plant and animal matter. They are categorized into humin, humic acid, and fulvic acid (FA), standing out due to their lower molecular weight and abundance of oxygen-rich functional groups [10]. These characteristics enable fulvic acids to pass through biological membrane micropores, unlike humic acids, which have larger molecular weights [14]. Fulvic acids also exhibit higher total acidity, a significant number of carboxyl groups, and superior adsorption and cation-exchange capacities, allowing them to act as natural chelators that facilitate the mobilization and transport of micronutrients and pass through cell membranes [15,16,17]. They support plant growth by enhancing photosynthesis, secretion of growth hormones, nutrient absorption, availability of pH, and boosting resistance to biotic and abiotic stresses [18]. Fulvic acids are a mixture of weak aliphatic and aromatic organic acids that are water-soluble across a broad pH range, including acidic, neutral, and alkaline conditions. The composition of fulvic acid is highly variable and is recognized as one of the most active carbon-based chelating agents.

Amino acids (AAs), often considered the fundamental building blocks of proteins, are nitrogen-containing compounds essential for protein synthesis [19]. They play a crucial role in metabolic processes by providing key enzymes that stimulate cell growth [20]. They are widely recognized as effective biostimulants that significantly enhance plant growth and development. Extensive research has demonstrated their multifaceted benefits across various plant species. AAs have been shown to improve fertilizer assimilation, optimize nutrient and water uptake, and enhance photosynthetic efficiency in numerous vegetable crops. These effects collectively contribute to increased flower production, improved fruit sets, and higher fruit yield, highlighting their vital role in boosting agricultural productivity [21].

Vermicompost is an organic fertilizer produced by earthworms by decomposing organic waste. It consists of the nutrient-rich excreta of earthworms, which enhances soil health and fertility [22]. This natural process transforms organic matter enriched with essential macro and micronutrients, beneficial microorganisms such as bacteria, fungi, and actinomycetes, and growth hormones like auxins, cytokinins, and gibberellins [23]. Vermicompost has been proven to act as both a growth promoter and a natural protector for crop plants [24]. These properties make it a growth promoter and a natural protector for crop plants. Vermicompost also improves soil structure, water retention, and aeration while gradually releasing nutrients to plants, making it an eco-friendly and sustainable biostimulant [25]. Vermicompost has emerged as a highly effective and promising biostimulant, demonstrating remarkable success in traditional soil-based cultivation and soilless techniques for vegetable production in greenhouses [26].

The extended cultivation period of iceberg lettuce, which is 2–3 times longer than that of Batavia-type lettuce due to the requirement for head formation, highlights the importance of improving cultivation practices. Faster growth is particularly crucial in soilless hydroponic water culture systems as it allows for increased production cycles per year. To address this, our study investigates using biostimulants to enhance plant nutrition, growth, quality, and yield. Although several studies have investigated the mentioned biostimulants, to our knowledge, no studies have systematically explored their application in soilless floating water culture systems for cultivating iceberg lettuce. By addressing this gap, the study contributes to advancing sustainable agricultural practices and explores the potential of biostimulants as innovative solutions for optimizing crop production in soilless systems.

This study evaluates the effects of fulvic acid, vermicompost, and amino acid biostimulants on the growth, yield, and quality of iceberg lettuce (*Lactuca sativa* var. *capitata*) grown in a hydroponic floating culture system. The research is based on the hypothesis that these eco-friendly biostimulants, applied individually or in combination, can synergistically enhance plant performance.

## 2. Results

### 2.1. Iceberg Lettuce Yield

The application of biostimulants significantly influenced iceberg lettuce yield in the first trial (Figure 1). The highest yield was observed with the application of vermicompost at 2 mL (VC 2 mL), reaching 12.58 kg m^−^^2^, corresponding to a 13.96% increase compared to the control treatment (11.04 kg m^−2^). Fulvic acid at 40 ppm (FA 40) also resulted in a notable yield increase, achieving 12.12 kg m^−2^, corresponding to a 9.78% increase compared to the control. However, no significant difference was observed between FA 40 and VC 2 mL. Other treatments, including FA 80, amino acid applications (AA 75 and AA 100), and vermicompost at 1 mL (VC 1 mL), did not significantly increase compared to the control. VC 1 mL exhibited a 4.35% increase, which, although higher than the control, was not statistically significant. In the first experiment, VC 2 mL and FA 40 demonstrated the most promising results for enhancing iceberg lettuce yield under experimental conditions.

The application of biostimulants significantly influenced iceberg lettuce yield during the second trial (Figure 2). Among the treatments, FA 40 + VC 2 mL exhibited the highest total yield, representing a 17.94% increase compared to the control. Similarly, FA 40 + AA 100 achieved a yield corresponding to a 17.11% increase relative to the control. The yields observed in FA 40 + AA 100 (17.16 kg m^−2^) and FA 40 + AA 100 + VC 2 mL (17.04 kg m^−2^) were statistically similar. The combination treatment of AA 100 + VC 2 mL produced a moderate yield, reflecting a 9.42% increase over the control. In contrast, the treatment FA 40 + AA 100 + VC 2 mL resulted in the lowest yield, indicating a 20.14% reduction compared to the control (14.55 kg m^−2^). Other treatments, such as AA 100 + VC 2 mL, yielded 15.92 kg m^−2^ and significantly differed from the control and the higher-yielding treatments.

### 2.2. Iceberg Lettuce Growth Parameters in First Experiment

Plant Height: The FA 80 led to a 6.3% increase in plant height compared to the control, with FA 40 showing a 5.6% improvement. VC 2 mL also contributed positively, achieving a 2.8% increase, indicating the beneficial effects of fulvic acid and vermicompost on plant height (Table 1).

Perimeter Dimension: FA 40 exhibited a slight advantage with a 0.6% increase over the control, and FA 80 displayed a 0.4% improvement.

Head Firmness: AA 75 was the only one to increase head firmness, achieving a 4.5% improvement compared to the control. All other treatments were lower than the control, highlighting AA 75 as the most effective parameter enhancement.

Root Weight: VC 2 mL achieved a notable 26% increase in root weight over the control, while AA 100 provided a 22% improvement and VC 1 mL yielded a 20.2% increase. These results underscore the efficacy of vermicompost and amino acid in promoting root biomass accumulation.

Leaf Fresh Weight: The VC 2 mL resulted in the highest leaf fresh weight, with a 14% increase compared to the control. FA 40 also showed a notable improvement, with a 10% increase, while AA 100 demonstrated a 3% rise (Table 2). These results suggest that vermicompost and fulvic acid applications positively impacted leaf fresh weight.

Leaf Area: AA 75 achieved the largest leaf area, showing a substantial 58% increase over the control. AA 100 followed with a 41% increase, and VC 2 mL exhibited a 45% improvement.

Number of Leaves: AA 100 application led to the highest leaf count, with a 20% increase relative to the control. VC 2 mL and VC 1 mL also showed positive impacts, with increases of 12% and 11.5%, respectively. This suggests that amino acids and vermicompost applications effectively enhance leaf production.

Dry Matter: In comparison to the control, dry matter content increased by 7.1%, 5.6%, 5.1%, 4.4%, 3.3%, and 1.4%, respectively, for the VC 2 mL, VC 1 mL, AA 100, FA 80, FA 40, and AA 75 treatments, although these increases were not statistically significant.

Chlorophyll Content (SPAD): The AA 100 demonstrated the highest chlorophyll content, increasing SPAD values by 31% relative to the control. VC 2 mL and FA 40 followed with 12% and 9% improvements, respectively, indicating that amino acid and vermicompost contributed positively to chlorophyll concentration.

### 2.3. Iceberg Lettuce Growth Parameters in Second Experiment

Plant height: The combined FA 40 + AA 100 yielded the highest plant height, with an increase of 6.4% compared to the control. FA 40 + VC 2 mL was closely followed, achieving a 6.2% increase, and AA 100 + VC 2 mL showed a 1.3% improvement over the control. (Table 3).

Perimeter: FA 40 + AA 100 demonstrated a 17% increase in perimeter dimension over the control, while AA 100 + VC 2 mL achieved a 10.2% increase. The FA 40 + VC 2 mL combination showed an 8.5% enhancement compared to the control.

Head firmness: In terms of head firmness, the FA 40 + AA 100 + VC 2 mL substantially increased, providing a 134.2% improvement over the control, making it the most effective treatment in this parameter. Other treatments, including FA 40 + AA 100, FA 40 + VC 2 mL, and AA 100 + VC 2 mL, demonstrated values below the control.

Root weight: FA 40 + VC 2 mL yielded the highest root weight, with a 33.9% increase compared to the control. The AA 100 + VC 2 mL combination also showed a substantial increase of 22.7%, and FA 40 + AA 100 demonstrated a 13.8% enhancement.

Leaf fresh weight: In the second trial, the combinations led to a higher increase in leaf fresh weight than in the first trial (Table 4). Specifically, the FA 40 + VC 2 mL and FA 40 + AA 100 produced 17–18% higher leaf fresh weight than the control.

Leaf area: The FA 40 + VC 2 ml treatment increased leaf area by 24.1%, while the FA 40 + AA 100 treatment resulted in a 9.7% increase compared to the control.

Number of leaves: The AA 100 + VC 2 mL combination yielded the highest leaf count, with a 9.6% increase compared to the control. This increase is more significant than observed in the first trial, suggesting that combinations were more supportive of leaf production than single biostimulant applications.

Dry matter: For dry matter content, the most substantial increases were observed in the FA 40 + AA 100 + VC 2 mL and AA 100 + VC 2 mL treatments, with enhancements of 43.8% and 35.9%, respectively, compared to the control.

Chlorophyll content (SPAD): Chlorophyll content did not show statistically significant differences among treatments.

### 2.4. Macro-Nutrient Profiles of Iceberg Lettuce

In the first trial of iceberg lettuce cultivation under soilless culture, significant variations in macro-nutrient concentrations were observed among treatments with different biostimulants (*p* < 0.05) (Table 5). Nitrogen concentration was highest in VC 2 mL, showing a substantial increase of 37.4% compared to the control. VC 1 mL and FA 40 followed, with increases of 23.0% and 15.7%, respectively. Phosphorus levels reached their maximum with VC 2 mL, surpassing the control by 19.2%. FA 80 showed a minor increase of 11.5%, while FA 40 minimally contributed to phosphorus enhancement. Potassium concentration peaked with FA 80, achieving a rise of 57.6% over the control. Strong performances were also observed with VC 2 mL and AA 75, which recorded increases of 45.4% and 42.4%, respectively. The highest calcium content was recorded with FA 40, representing an improvement of 18.0% relative to the control. FA 80 followed closely with a 15.0% increase, and AA 75 contributed moderately to calcium enhancement. Magnesium concentrations exhibited the most significant increase with AA 75, rising by 144.3% compared to the control. FA 80 and FA 40 also demonstrated notable improvements, with increases of 102.5% and 69.6%, respectively.

In the second trial of lettuce cultivation under soilless conditions, the macronutrient concentrations showed significant differences among treatments, with some combinations outperforming others (Table 6). Nitrogen concentration reached its highest levels in FA 40 + VC 2 mL, demonstrating a considerable improvement of 37.4% over the control. FA 40 + AA 100 and AA 100 + VC 2 mL followed closely, showing similar enhancements in nitrogen levels. Phosphorus concentration was most elevated in FA 40 + VC 2 mL, achieving an increase of 69.2% compared to the control. FA 40 + AA 100 ranked second with a 65.4% increase, while the AA 100 + VC 2 mL combination also performed well, with a moderate enhancement. Although statistically not significant, potassium concentration peaked in FA 40 + VC 2 mL, which improved by 43.5% compared to the control. AA 100 + VC 2 mL followed with an increase of 43.2%, and FA 40 + AA 100 ranked third with a 33.2% improvement. FA 40 + VC 2 mL recorded the highest concentration for calcium, reflecting an improvement of 68.6% over the control. FA 40 + AA 100 followed with a 50.0% increase, while AA 100 + VC 2 mL showed an enhancement of 48.5%. Magnesium levels were highest in AA 100 + VC 2 mL, marking a 92.4% increase over the control. FA 40 + AA 100 contributed a 69.6% improvement, while FA 40 + VC 2 mL achieved a rise of 65.8%.

### 2.5. Antioxidant Contents of Iceberg Lettuce

In the first trial of iceberg lettuce cultivation under soilless culture, significant differences were observed in antioxidant parameters (Table 7). Vitamin C content was highest in VC 1 mL, showing a 33.9% increase compared to the control. VC 2 mL ranked second with a 28.1% improvement, followed by AA 100, which increased Vitamin C levels by 25.6%. The application of biostimulants resulted in an increase in total phenol content compared to the control for the following treatments: FA 40 (25.34%), AA 100 (8.96%), and VC 2 mL (7.54%). Other treatments did not show an increase in total phenol content. In terms of total flavonoid content, the treatments that exhibited an increase compared to the control were FA 40 (60.64%), AA 100 (31.38%), VC 2 mL (29.26%), and AA 75 (15.96%). Other treatments did not increase total flavonoid content.

In the second trial in the spring, the combination treatments demonstrated notable increases in antioxidant content compared to the control (Table 8). For vitamin C, the highest increase was observed with the FA 40 + AA 100 + VC 2 mL treatment, which showed a 17.16% improvement, followed by FA 40 + AA 100 (15.06%), FA 40 + VC 2 mL (9.11%), and AA 100 + VC 2 mL (4.20%). Regarding total phenol content, FA 40 + AA 100 + VC 2 mL exhibited the most significant increase at 52.54%, while FA 40 + AA 100 and AA 100 + VC 2 mL also showed substantial increases of 40.50% and 43.06%, respectively. The FA 40 + VC 2 mL treatment resulted in a 22.28% increase. For total flavonoid content, FA 40 + AA 100 + VC 2 mL again led with a 52.38% increase, followed by FA 40 + AA 100 at 46.32%, AA 100 + VC 2 mL at 42.42%, and FA 40 + VC 2 mL at 35.50%.

### 2.6. Leaf Quality Parameters of BRIX, Acidity, pH, and EC Values of Iceberg Lettuce

In the first trial of iceberg lettuce cultivation under soilless conditions, the effects of biostimulants on brix, acidity, pH, and EC values were evaluated. In both trials, the Brix values in lettuce leaves did not show any statistically significant differences among the treatments. Among these parameters, pH showed statistically significant differences (*p* < 0.05), while other parameters exhibited no variation (Table 9). pH levels showed significant variation, with the highest values recorded in FA 80, FA 40, and the control, all maintaining statistically similar levels. The lowest pH levels were observed in the VC 1 mL and AA 100, significantly reduced compared to the control.

In the second trial of iceberg lettuce under soilless culture, statistically significant differences were observed for pH (*p* < 0.05) and EC (*p* < 0.05). At the same time, brix and acidity values did not show significant variations (Table 10). pH levels demonstrated significant differences among treatments. The highest pH values were recorded in FA 40 + VC 2 mL and FA 40 + AA 100 + VC 2 mL, maintaining slightly higher levels than the control. Conversely, FA 40 + AA 100 and AA 100 + VC 2 mL recorded lower pH values, indicating a notable reduction. Electrical conductivity values were significantly affected by the treatments. In comparison to the control, the highest increases in EC values were observed with the FA 80, FA 40, and AA 100 treatments, showing increases of 51.5%, 47.4%, and 44.0%, respectively. These results indicate that fulvic acid and amino acid applications significantly enhanced the EC levels of the growing medium. The control recorded one of the lowest EC values among all treatments.

### 2.7. Nitrate Content in Iceberg Lettuce

In the first trial of iceberg lettuce cultivation, the nitrate reduction capacity of biostimulants was evident, with specific treatments significantly decreasing nitrate accumulation in leaf tissues (Figure 3). The lowest nitrate content was recorded in AA 100 (690 mg/kg), demonstrating a remarkable reduction of 39.1% compared to the control (1133 mg/kg). This highlights AA 100 as the most effective biostimulant in mitigating nitrate accumulation, making it highly desirable for enhancing lettuce quality. Among the vermicompost, VC 2 mL (1078 mg/kg) and VC 1 mL (1085 mg/kg) achieved moderate reductions of 4.9% and 4.2%, respectively, compared to the control. While these reductions are less pronounced, they still reflect the potential of vermicompost to limit nitrate levels. Conversely, FA-based, including FA 80 (1238 mg/kg), FA 40 (1229 mg/kg), and AA 75 (1189 mg/kg), showed increased nitrate content, with levels exceeding the control. These results suggest that FA-based are less effective or may even promote nitrate accumulation under the conditions tested. Overall, the findings demonstrate that the nitrate-reducing capacity of biostimulants varies significantly, with AA 100 emerging as the most effective treatment for reducing nitrate content in lettuce leaves, a highly desirable outcome for improved quality and safety.

In the second trial of iceberg lettuce cultivation, biostimulants demonstrated varying capacities to reduce nitrate content in leaf tissues, highlighting their potential to enhance lettuce quality (Figure 4). The lowest nitrate content was recorded in FA 40 + AA 100 + VC 2 mL (275 mg/kg), representing a significant reduction of 56.2% compared to the control (628 mg/kg). This treatment emerged as the most effective in mitigating nitrate accumulation, underscoring the synergistic effects of combining multiple biostimulants. AA 100 + VC 2 mL also showed a substantial reduction in nitrate content, with a 44.9% decrease compared to the control. This highlights its effectiveness in reducing nitrate levels while maintaining leaf quality. In contrast, FA 40 + AA 100 (646 mg/kg) and FA 40 + VC 2 mL (596 mg/kg) recorded nitrate contents similar to or slightly higher than the control, indicating limited or no nitrate-reducing capacity under the tested conditions. The results demonstrate that combining biostimulants reduces nitrate levels in lettuce leaves. FA 40 + AA 100 + VC 2 mL is particularly effective, making it a desirable option for achieving high-quality lettuce with reduced nitrate content.

## 3. Discussion

The exogenous application of biostimulants, including amino acids, vermicompost, and fulvic acid, represents an innovative and eco-friendly agricultural approach that complements mineral fertilization, enhances plant productivity, promotes growth, increases yield, and improves product quality.

### 3.1. Effects of Biostimulants on Plant Growth Parameters and Iceberg Lettuce Yield

The findings in this study reveal that biostimulants enhanced lettuce performance, promoting growth parameters, chlorophyll levels, and nutrient absorption. Vermicompost achieved the most substantial increases in root biomass and leaf area. In contrast, combinations of amino acids and fulvic acid effectively reduced nitrate content while increasing antioxidants and dry matter accumulation. The combined application of FA 40 + VC 2 mL resulted in an 18.12% increase in iceberg lettuce yield compared to the control. At the same time, the combination of FA 40 + AA 100 led to a 17.4% increase.

Biostimulants are rich in growth regulators, including humic substances, cytokinins, and auxins, contributing to increases in root and leaf biomass, plant height, leaf number, and shoot and root dry weight. Canellas et al. [27] and Taha et al. [16] reported that the application of fulvic acid significantly enhanced lettuce growth parameters compared to the control. The foliar application at 3.1 g m^−2^ resulted in a 126% increase in fresh weight, 127% in dry weight, and 30% in plant height. Similarly, irrigation application at 3.1 g m^−2^ increased by 105% in fresh weight, 103% in dry weight, and 14% in plant height. These results demonstrate the effectiveness of fulvic acid, mainly through foliar application, in promoting lettuce growth. Lüdtke et al. [28] showed that the application of FA + NPK significantly improved lettuce growth compared to the control. By day 48, the number of leaves increased by 51%, plant diameter by 40%, plant height by 33%, and root length by 45%. Additionally, the aerial fresh mass of lettuce reached approximately 280 g/plant under FA + NPK treatment, representing a 180% increase compared to the control. These findings highlight the potential of FA + NPK to enhance overall lettuce growth, contributing to both shoot and root development and biomass production.

Fulvic acid application plants increased the net photosynthetic rate and chlorophyll fluorescence characteristics, including initial fluorescence (Fo), steady-state fluorescence (Fs), maximum fluorescence (Fm), variable fluorescence (Fv), photochemical quenching coefficient (qP), and the effective photochemical quantum yield of PSII (Fv’/Fm’) and promoted photosynthesis [29]. In addition, fulvic acids enhance photosynthesis by modulating signaling pathways associated with plant hormones. They improve essential physiological parameters, including transpiration rate, stomatal conductance, water-use efficiency, intercellular CO_2_ concentration, chlorophyll content, and relative water content. These findings underscore the significant potential of fulvic acids to enhance plant growth and productivity by optimizing critical physiological processes [30].

Al-Karaki and Othman [31] demonstrated that in soilless cultivation using a tuff-zeolite medium, foliar application of an amino acid biostimulant at a concentration of 4 mL/L significantly increased yields, with a 26% improvement in iceberg lettuce and a 50% increase in romaine lettuce compared to the control. Amino acids are building blocks for protein biosynthesis and precursors for various biosynthetic pathways and key regulators in signaling processes and stress responses [32]. Their role in promoting a more robust root system enhances water and nutrient uptake efficiency, ultimately driving better plant growth and yield [33]. Haghighi et al. [21] further reported that amino acid application improved the fresh and dry weights of cabbage shoots and the fresh weight of roots by enhancing key physiological processes such as photosynthesis, transpiration, and stomatal conductance. These improvements were linked to enhanced nitrogen uptake and assimilation, contributing to more efficient CO_2_ assimilation and the effective transport of soluble sugars via the phloem to sink tissues. The biostimulatory effects of amino acids are also associated with activating critical enzymes, such as nitrate reductase and glutamine synthase, which are pivotal in nitrogen metabolism [34]. These findings underscore the multifaceted benefits of amino acid biostimulants in improving plant growth and productivity, particularly under soilless culture systems.

According to Yassen et al. [34], the application of 0.95 kg m^−2^ o fvermicompost combined with 150 mL L^−1^ of vermiwash, a liquid extract derived from vermicomposting processes, significantly improved lettuce yield, increasing it from 2.48 kg m^−2^ in the control to 4.72 kg m^−2^, a remarkable 91% increase. Similarly, Kilic and Saracoglu [35] reported substantial yield enhancements with vermicompost application, achieving a 121% increase in iceberg lettuce and an 184% increase in Lollo Rosso lettuce. These findings emphasize the efficacy of vermicompost and its combination with vermiwash in enhancing lettuce productivity. The beneficial effects of vermicompost on plant growth are attributed to its role in improving photosynthesis and chlorophyll fluorescence. Zou et al. [36] demonstrated that vermicompost significantly increased key physiological parameters, including net photosynthetic rate, stomatal conductance, transpiration rate, and intercellular CO_2_ concentration. Furthermore, chlorophyll fluorescence traits, such as maximum fluorescence (Fm), maximum photochemical efficiency (Fv/Fm), and potential photochemical activity (Fv/F_0_), were also enhanced, contributing to improved plant growth and yield [37].

The positive impacts of vermicompost are linked to its high nutrient content and bioactive composition. Vermicompost contains essential macronutrients (N, P, K, Ca, Mg, and S) and micronutrients (Fe, Mn, Zn, Cu, and B), which are critical for supporting root, shoot, and fruit development [38]. Additionally, it is enriched with beneficial microorganisms, vitamins, growth hormones, and enzymes such as proteases, amylases, cellulases, and chitinases, which promote plant development [23]. The microbial diversity in vermicompost, particularly bacterial and fungal populations, further enhances its growth-promoting properties [39]. Another key advantage of vermicompost is its abundance of plant growth regulators, including humic substances, cytokinins, and auxins. These compounds are likely responsible for the observed increases in leaf and root weights, leaf number, plant height, leaf area, and dry weight of shoots and roots [10]. Toor et al. [22] highlighted the effectiveness of vermicompost in enhancing lettuce growth, reporting a 100% increase in total chlorophyll content, a 118% increase in fresh weight, and a 20% improvement in plant height with the optimal application dose of 2%.

### 3.2. Effects of Biostimulants on Mineral Nutrient Contents

Biostimulants, particularly fulvic acid, amino acids, and vermicompost, play a significant role in enhancing the mineral content of plants. Biostimulants primarily stimulate root development, increasing the plant’s nutrient uptake capacity, as highlighted in [40]. According to Yassen et al. [34], the application of 0.95 kg m^−2^ vermicompost + 150 mL L^−1^ vermiwash significantly improved the N, P, K, Ca, and Mg contents in the head of lettuce compared to the control. The increases were 46%, 43%, 45%, 16%, and 62%, respectively. These results demonstrate the effectiveness of this optimal vermicompost dose in enhancing both the macronutrient and micronutrient composition of lettuce heads, contributing to their overall nutritional quality. A more robust root system enables the efficient absorption and transport of nutrients to plant tissues. Moreover, biostimulants like vermicompost enhance nutrient availability by providing organic compounds and beneficial microorganisms that facilitate nutrient uptake [41]. Additionally, biostimulants influence hormone balance and signaling pathways to support nutrient mobilization. Elevated cytokinin levels promote nutrient transport to source tissues, such as leaves, enhancing plant growth [42].

Al-Karaki and Othman [31] demonstrated that in the soilless culture of iceberg lettuce, the foliar application of an amino acid-based biostimulant led to notable increases in leaf mineral concentrations compared to the control. Specifically, for iceberg lettuce, nitrogen (N) content increased by approximately 29%, phosphorus (P) by 25%, potassium (K) by 22%, and magnesium (Mg) by 53% [31]. Lüdtke et al. [28] demonstrated that the application of FA + NPK significantly improved the macronutrient and micronutrient contents of lettuce compared to the control. N, P, K, Ca, and Mg contents increased by 11%, 14%, 67%, 15%, and 2%, respectively. These findings underscore the effectiveness of FA + NPK in enhancing plant nutrient levels, contributing to the nutritional quality of lettuce. Biostimulants also improve photosynthesis, supporting energy metabolism and facilitating active nutrient transport and ion uptake in roots [41]. Amino acids contribute to increased carbohydrate production, meeting the energy demands of roots and improving nutrient acquisition [43]. Vermicompost supports beneficial microbial populations, further enhancing the bioavailability of essential minerals [36].

### 3.3. Effects of Biostimulants on Iceberg Lettuce Quality and Antioxidant Contents

In this study, biostimulants, particularly fulvic acid, amino acids, and vermicompost, have proven effective in enhancing vitamin content, antioxidant capacity, phenolic compounds, and flavonoid levels in lettuce. Plants treated with amino acids exhibited higher phenolic content and antioxidant activity, which can be partially attributed to the localized beneficial effects of amino acids on plant processes. Amino acids are precursors to a wide range of secondary metabolites and are closely linked to synthesizing proteins, hormones, and carbohydrates [31]. Haghighi et al. [21] demonstrated that the exogenous application of AAs enhances the nutritional profile of cabbage, including increased levels of phenols, total protein, proline, and essential amino acids such as glutamic acid, glutamine, and asparagine, along with improved antioxidant capacity.

Vermicompost has enhanced key quality compounds in strawberries, including soluble sugar and vitamin C content [36]. Al Jaouni et al. [44] reported that vermicompost exhibited the highest levels of antioxidants and bioactive compounds, including phenolics, flavonoids, steroids, and ascorbate. Abdel–Baky et al. [30] demonstrated that applying fulvic acid significantly increased total carbohydrate content, crude protein, and essential minerals. Additionally, it elevated the levels of key amino acids such as arginine, lysine, phenylalanine, and tryptophan. The foliar application of fulvic acid significantly enhanced secondary metabolites, including total phenolic content, anthocyanins, and total carotenoids, underscoring its potential to improve quality attributes [14].

### 3.4. Effects of Biostimulants on Iceberg Lettuce Nitrate Content

During the second experimental period (February, March, and April), the light intensity inside the greenhouse ranged between 350 and 550 W·m^−2^, and the day length extended from 10.4 to 13.7 h. These conditions may have contributed to the observed reduction in nitrate levels, as increasing light intensity and extended day length play a critical role in nitrate metabolism in leafy vegetables [45]. In this study, applying the biostimulant combination FA 40 + AA 100 + VC 2 mL resulted in the highest reduction in nitrate content, decreasing levels by 56.2% (275 mg/kg) compared to the control (628 mg/kg). Similarly, AA 100 + VC 2 mL significantly reduced nitrate content by 44.9%. These findings align with the role of nitrate reductase (NR) in plant nitrogen metabolism, where NR catalyzes the reduction of nitrate to ammonium, a precursor for amino acid synthesis [46].

In the second spring trial, biostimulant applications significantly decreased nitrate content in lettuce leaf tissues. This is consistent with the findings of Coronel et al. [47], who reported that soilless lettuce cultivation enhances NR activity, making it a promising method for producing lettuce with reduced nitrate levels. Biostimulants promote plant growth by increasing amino acid concentrations, which are directly utilized in protein biosynthesis. Elevated carbohydrate levels in the leaves further support this process by enhancing nitrogen assimilation through the nitrate pathway. Carbohydrates provide the carbon skeletons required for converting reduced nitrate (ammonia) into amino acids, facilitating protein synthesis and overall growth [10]. Additionally, applying exogenous amino acids has reduced nitrate concentrations in cabbage leaves, suggesting that externally supplied amino acids may regulate nitrate uptake [21]. In our study, nitrate levels of both trials were significantly lower than the safety limits for human health. According to the European Commission regulation (EC Reg. No. 1258/2011), the maximum allowable concentration is 5000 mg kg^−1^ FW for lettuce grown in protected conditions under cover. Our research complies with this standard, as it was conducted between November and April [45].

## 4. Materials and Methods

The research was conducted in a glasshouse at the Department of Horticulture, Faculty of Agriculture, Çukurova University, during the winter and early spring seasons. This study aimed to assess the impact of biostimulants on iceberg lettuce cultivation within a floating culture soilless system (Figure 5). The plant material used was the “Bombola” iceberg lettuce variety, sourced from the seed company “Enza Zaden”. Biostimulants were applied to enhance plant growth, yield, and nutrient content in iceberg lettuce. To establish effective biostimulant dosages, we organized two distinct experimental trials. The first trial was conducted during the winter months (November, December, and January) for 74 days, while the second trial was conducted in the early spring months (February, March, and April) for 70 days in a glasshouse environment.

### 4.1. Biostimulants

Fulvic acid, amino acid, and vermicompost were the biostimulants [26]. The fulvic acid and amino acid products were obtained from “Köklü Group” and marketed under the brand names “Sacaka WS” and “Aminoset,” respectively. “Sacaka WS” has the following specifications: 80% organic matter content, 70% total (humic + fulvic) acid content, 70% fulvic acid content, and a pH range of 2–4. “Aminoset” contains 50% organic matter, 20% organic carbon, 4% organic nitrogen, 30% free amino acids, and a pH range of 3.1–5.1. The vermicompost, branded as “EkosolFarm”, 100% organic liquid vermicompost, was sourced from the “Ekosol Tarim” company. Vermicompost products are typically rich in organic matter (up to 70%), humic acid (up to 30%), and fulvic acid (up to 20%). They also contain essential macronutrients and micronutrients, such as nitrogen, phosphorus, potassium, calcium, magnesium, iron, zinc, enzymes, beneficial microorganisms, and natural growth hormones, which promote plant growth, resilience, and yield. The vermicompost used in this study was produced from agricultural waste and manure.

### 4.2. Experimental Conditions

The dimensions of the containers used in the hydroponic system were 73.6 × 46 × 20 cm (L × W × H), with lid-shaped plates measuring 105 cm × 55 cm (L × W). The spacing between rows of iceberg lettuce plants was maintained at 18.4 cm × 18.4 cm, resulting in a plant density of 29.54 plants m^−^^2^. Yield (kg/m^2^) was calculated based on the harvest of 10 iceberg lettuce plants grown in an area of 0.33856 m^2^.

The first trial included seven treatments: a control, two doses of fulvic acid (40 and 80 ppm), amino acid (75 and 100 ppm), and vermicompost (liquid; 1 and 2 mL L^−1^). The second trial was established using combinations of biostimulant doses that demonstrated the best performance in the first trial. In the second experiment, five treatments were applied, including a control, fulvic acid at 40 ppm + amino acid at 100 ppm, fulvic acid at 40 ppm + vermicompost at 2 mL L^−1^, amino acid at 100 ppm + vermicompost at 2 mL L^−1^, and a combination of fulvic acid at 40 ppm + amino acid at 100 ppm + vermicompost at 2 mL L^−1^.

The nutrient solution was prepared to supply essential nutrients for plant growth [48]. The elemental composition of the nutrient solution for lettuce plants included (in mg L^−1^): nitrogen (230), phosphorus (50), potassium (320), calcium (220), magnesium (60), iron (4.0), zinc (0.50), boron (0.51), copper (0.23), molybdenum (0.18), and manganese (0.78).

The nutrient solution’s electrical conductivity (EC) level was gradually increased from 1.5 to 2.4 dS·m^−1^ as the iceberg lettuce plants progressed through their phenological growth stages. This adjustment was made to meet the increasing mineral nutrient requirements of the plants, particularly during head formation. The nutrient solution was replaced with a fresh nutrient solution every 10 days. The pH values of the nutrient solution were maintained between 6.0 and 6.5. The nutrient solution was continuously aerated, with oxygen levels fluctuating between 6 and 8 ppm throughout the experiments. During the first trial, the greenhouse temperatures ranged between 15 °C and 21 °C, while in the second trial, they ranged between 17 °C and 24 °C. During the first experimental period (November, December, and January), the light intensity inside the greenhouse ranged between 250 and 350 W·m^−2^, and the day length varied from 9.5 to 10.6 h. In the second experimental period (February, March, and April), the light intensity inside the greenhouse ranged between 350 and 550 W·m^−2^, while the day length extended from 10.4 to 13.7 h.

The “Bombola” iceberg lettuce variety reached the harvest stage 74 days after transplanting in the first experiment, while the second harvest occurred 70 days post-transplanting. Parameters measured across both experiments included total yield (kg m^−2^), plant height (cm), perimeter length (cm), head firmness (kg cm^−^³), root weight (g), leaf fresh weight (g plant^−1^), leaf area (cm^2^ plant^−1^), leaf count (leaves plant^−1^), dry matter content (%), chlorophyll content (SPAD), and dry weight (g plant^−1^). Leaf samples were analyzed for nitrogen (N), phosphorus (P), potassium (K), magnesium (Mg), calcium (Ca), iron (Fe), manganese (Mn), copper (Cu), and zinc (Zn) concentrations. Nitrate concentration (ppm), vitamin C (L-ascorbic acid), total phenolic, and flavonoid contents were also measured. For lettuce leaves, total soluble solids (TSS, %), titratable acidity (%), pH, and electrical conductivity (EC, mS cm^−1^) were also determined.

### 4.3. Measurements of Plant Growth Parameters

At harvest time, the yield (kg m^−2^) was computed by summing the total weight of 10 plants within each replicate, with leaf counts also documented. Lettuce head firmness was evaluated using a digital penetrometer (Bareiss HPE-III-Fff, ABQ Industrial, Woodlands, TX, USA), with results recorded in kilograms [49]. Leaf area per plant (cm^−2^) was measured using a leaf area meter (Li-3100, LICOR, Lincoln, NE, USA). After harvest, each plant was weighed on a digital scale to determine fresh leaf weight per plant (g). Chlorophyll content in the leaves was measured by a SPAD chlorophyll meter (Minolta 502, Tokyo, Japan). The harvested leaves’ luminosity (L) and chromaticity coordinates [a (red–green) and b (blue–yellow)] were recorded digitally via a handheld color spectrophotometer (HunterLab, Reston, VA, USA) [50]. Following this, fresh leaves were dried at 65 °C for 48 h to obtain dry weight per plant.

### 4.4. Determination of Total Soluble Solids (TSS), pH, and Electrical Conductivity (EC) in Lettuce Leaves

The juice extracted from lettuce leaves determined total soluble solids (TSS), pH, and electrical conductivity (EC). TSS was measured with a digital refractometer (Atago PR-101, Tokyo, Japan). At the same time, pH and EC values were assessed using a pH and EC meter (WTW pH/Cond 3320, Weilheim, Germany) [11].

### 4.5. Determination of Nitrate Concentration

Nitrate concentration in lettuce leaves was assessed using the salicylic acid method outlined by Cataldo et al. (1975), with measurements taken colorimetrically at 410 nm. Results were reported in milligrams per kilogram of fresh weight [50].

### 4.6. Determination of Ascorbic Acid Content (Vitamin C)

The ascorbic acid content in lettuce leaves was analyzed following a modified method by [51]. Lettuce leaves were processed using a juicer for extraction. Five milliliters of the extracted juice were mixed with 45 milliliters of 0.4% oxalic acid and then filtered. From the filtered solution, one milliliter was combined with nine milliliters of 2,6-dichlorophenolindophenol and thoroughly mixed. The samples were then measured at 520 nm using a UV spectrophotometer (UV-1700 PharmoSpec Shimadzu, Kyoto, Japan).

### 4.7. Determination of Total Phenolic and Flavonoid Compounds

The total phenolic content in lettuce leaves was measured with a modified spectrophotometric method based on Spanos and Wrolstad [52]. Absorbance readings were taken at 765 nm using a spectrophotometer (UV-1700 PharmoSpec Shimadzu, Kyoto, Japan) with a gallic acid calibration curve for quantification. Total flavonoid content was assessed at 415 nm, following the method by Quettier–Deleu et al. [53] and quantified using a calibration curve prepared with rutin.

### 4.8. Leaf Mineral Nutrient Analysis

Leaf mineral nutrient analysis was performed by sampling one-quarter of three plants from each replicate at harvest, using the method outlined by Dasgan et al. [49]. The analysis included macronutrients (N, P, K, Mg, Ca) and micronutrients (Fe, Mn, Cu, Zn). Lettuce leaf samples were thoroughly cleaned, rinsed three times with distilled water to avoid contamination, and dried at 65 °C for 48 h using oven-drying. The dried leaves were ground to a 40-mesh particle size using a leaf grinder. For K, Ca, Mg, Na, Fe, Mn, Zn, and Cu analysis, 0.2 g of the ground sample were incinerated at 550 °C for 5 h, and the resulting ash was dissolved in 3.3% HCl (*v*/*v*) and filtered [49]. Potassium, calcium, magnesium, and sodium were analyzed in emission mode, while iron, manganese, zinc, and copper were analyzed in absorbance mode with an Atomic Absorption Spectrophotometer (Varian FS220, Palo Alto, CA, USA). Leaf nitrogen and phosphorus were determined using the Kjeldahl and Barton methods, respectively [54].

### 4.9. Statistical Analysis

This study conducted both experiments using a randomized block design with four replicates, each containing 10 plants. The data obtained were analyzed using the JMP statistical software 13 version, and the means were compared using the Tukey test at a 5% significance level.

## 5. Conclusions

This study demonstrates the significant potential of biostimulants such as fulvic acid, amino acids, and vermicompost in enhancing the growth, yield, and nutritional quality of soilless-grown iceberg lettuce. Applying these eco-friendly biostimulants, individually and in combination, improved key plant parameters, including biomass production, nutrient uptake, and antioxidant content, while reducing nitrate levels in lettuce leaves. Notably, FA 40 + VC 2 mL achieved the highest yield (17.15 kg/m^2^), an 18.2% increase over the control, followed closely by FA 40 + AA 100 (17.03 kg/m^2^, 17.4% increase). Both combinations significantly enhanced lettuce yield. These findings underscore the value of biostimulants as sustainable solutions for modern agriculture, reducing dependency on conventional fertilizers and mitigating environmental impacts. Biostimulants are a promising approach for advancing sustainable soilless culture systems and addressing the growing global demand for high-quality, nutrient-rich vegetables.

Biostimulants provide sustainable solutions for soilless systems by enhancing nutrient-use efficiency and amplifying the effectiveness of conventional fertilizers. Future research should aim to optimize biostimulant application strategies by exploring novel biostimulants, innovative combinations, and optimal dosing regimens to maximize their efficacy and investigate their long-term benefits for sustainable soilless culture production.

## Figures and Tables

**Figure 1 plants-14-00609-f001:**
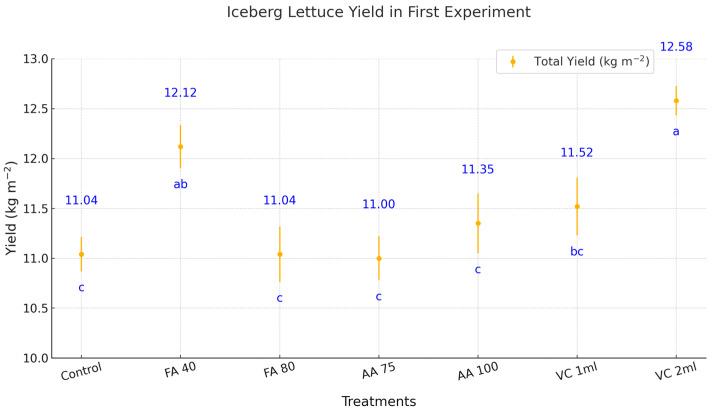
The effect of biostimulants on iceberg lettuce yield in the first winter trial. FA = Fulvic acid, AA = Aminoacid, VC = Vermicompost. No significant difference was observed between means represented by the same letter on the point (*p* < 0.05).

**Figure 2 plants-14-00609-f002:**
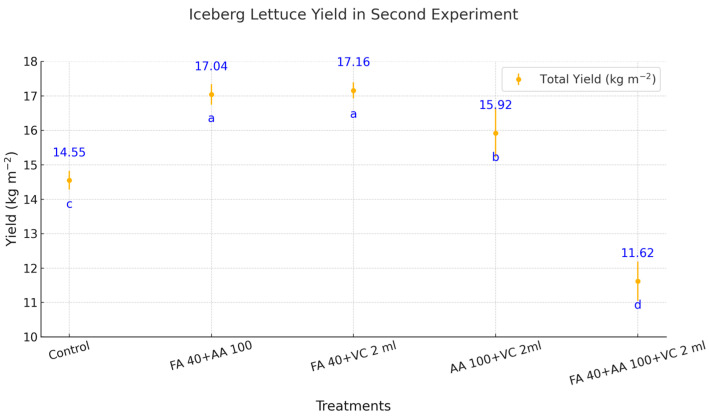
The effect of biostimulants on iceberg lettuce yield in the second spring trial. FA = Fulvic acid, AA = Aminoacid, VC = Vermicompost. No significant difference was observed between means represented by the same letter on the point (*p* < 0.05).

**Figure 3 plants-14-00609-f003:**
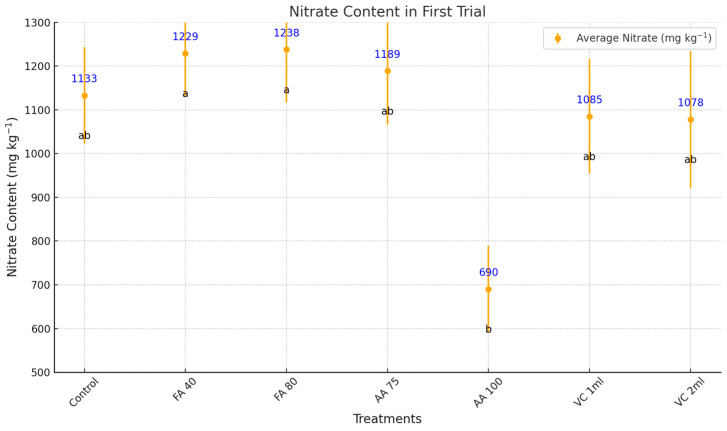
Effects of biostimulants on nitrate content in soilless-grown iceberg lettuce in first winter experiment. FA = Fulvic acid, AA = Aminoacid, VC = Vermicompost. No significant difference was observed between means represented by the same letter on the point (*p* < 0.05).

**Figure 4 plants-14-00609-f004:**
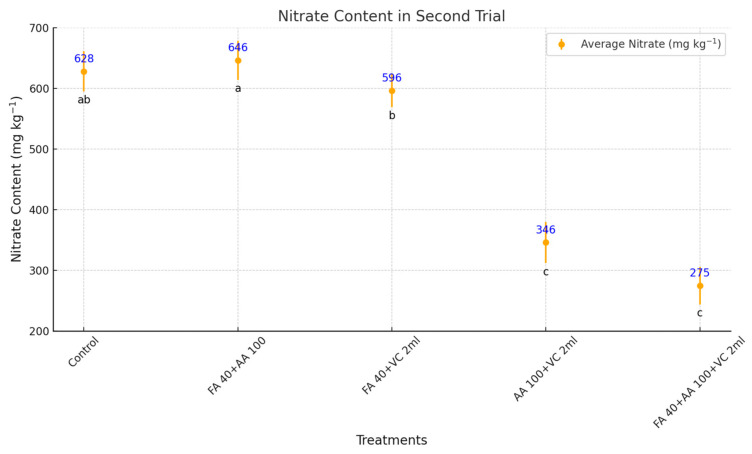
Effects of biostimulants on nitrate content in soilless-grown iceberg lettuce in second spring experiment. FA = Fulvic acid, AA = Aminoacid, VC = Vermicompost. No significant difference was observed between means represented by the same letter on the point (*p* < 0.05).

**Figure 5 plants-14-00609-f005:**
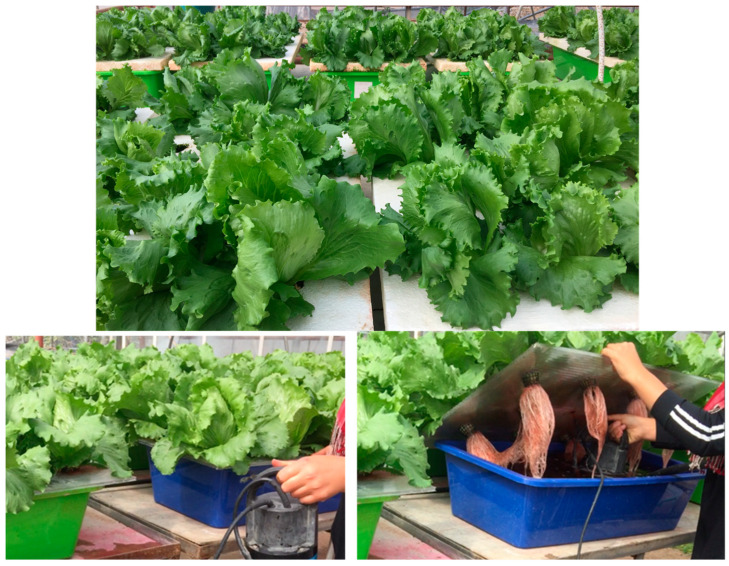
Iceberg lettuce grown in a floating hydroponic system, showing plant growth, system setup, and root development.

**Table 1 plants-14-00609-t001:** The effects of biostimulants on growth parameters of iceberg lettuce in the first trial-I.

Treatments	Plant Height (cm)	Perimeter (cm)	$\mathrm{Head}\;\mathrm{Firmness}\;(\mathrm{kg}\;{cm}^{3}$)	Root Weight (g)
Control	38.85 abc	44.85 a	8.24 ab	45.40
FA 40	41.05 a	45.10 a	6.88 abc	50.00
FA 80	41.30 a	45.05 a	7.73 ab	48.50
AA 75	37.30 bc	40.85 b	8.61 a	44.70
AA 100	33.60 d	43.55 ab	6.75 bc	55.40
VC 1 mL	35.55 cd	41.75 ab	5.82 c	54.60
VC 2 mL	39.95 ab	44.00 ab	6.67 bc	57.20
*p*	<0.0001	0.0011	<0.0001	0.1211
${Tukey}_{0.05}$	3.516	3.575	1.733	NS

FA = Fulvic acid, AA = Aminoacid, VC = Vermicompost. No significant difference was observed between means sharing the same letter within the same column (*p* < 0.05). NS: Not Significant.

**Table 2 plants-14-00609-t002:** The effects of biostimulants on growth parameters of iceberg lettuce in the first trial-II.

Treatments	Leaf Fresh Weight (g Plant^−1^)	Leaf Area (cm^2^ Plant^−1^)	Number of Leaves per Plant	Dry Matter (%)	Chlorophyll (SPAD)
Control	372 d	4793 b	33.15 c	7.31	16.44 ab
FA 40	409 b	5952 ab	34.25 bc	7.55	17.86 ab
FA 80	372 d	5791 ab	34.80 bc	7.63	17.74 ab
AA 75	371 d	7585 a	33.05 c	7.41	15.04 b
AA 100	383 c	6748 ab	39.85 a	7.68	21.61 a
VC 1 mL	388 c	6071 ab	36.95 ab	7.72	17.53 ab
VC 2 mL	424 a	692 ab	37.15 ab	7.83	18.46 ab
*p*	<0.0001	0.0398	<0.0001	0.1951	0.0215
${Tukey}_{0.05}$	5.719	2515	3.654	NS	5.329

FA = Fulvic acid, AA = Aminoacid, VC = Vermicompost. NS: Non-significant. No significant difference was observed between means sharing the same letter within the same column (*p* < 0.05). NS: Not Significant.

**Table 3 plants-14-00609-t003:** The effects of biostimulants on growth parameters of iceberg lettuce in the second trial-I.

Treatments	Plant Height (cm)	Perimeter (cm)	Head Firmness (kg·cm^−3^)	Root Weight (g)
Control	29.42 b	38.85 bc	9.47 b	70.90 c
FA 40 + AA 100	31.30 a	45.45 a	8.96 b	80.70 b
FA 40 + VC 2 mL	31.25 a	42.15 ab	8.75 b	95.00 a
AA 100 + VC 2 mL	29.80 ab	42.80 ab	8.85 b	87.00 ab
FA 40 + AA 100 + VC 2 mL	29.20 b	37.55 c	22.17 a	72.10 c
*p*	0.0008	<0.0001	<0.0001	<0.0001
${Tukey}_{0.05}$	1.743	4.1311	1.744	8.294

FA = Fulvic acid, AA = Aminoacid, VC = Vermicompost. No significant difference was observed between means sharing the same letter within the same column (*p* < 0.05).

**Table 4 plants-14-00609-t004:** The effects of biostimulants on growth parameters of iceberg lettuce in the second trial.

Treatments	Leaf Fresh Weight (g Plant^−1^)	Leaf Area (cm^2^ Plant^−1^)	Number of Leaves per Plant	Dry Matter (%)	Chlorophyll (SPAD)
Control	490 b	10,943 bc	33.90 ab	9.92 c	20.45
FA 40 + AA 100	574 a	12,005 b	35.70 ab	11.16 bc	19.67
FA 40 + VC 2 mL	579 a	13,580 a	35.60 ab	11.04 bc	17.80
AA 100 + VC 2 mL	537 ab	10,492 c	37.15 a	13.48 ab	20.02
FA40 + AA100 + VC 2 mL	392 c	10,308 c	32.20 b	14.26 a	19.97
*p*	<0.0001	<0.0001	0.0025	<0.0001	0.3022
${Tukey}_{0.05}$	82.545	1389	3.549	1.916	NS

FA = Fulvic acid, AA = Aminoacid, VC = Vermicompost. NS: Non-significant. No significant difference was observed between means sharing the same letter within the same column (*p* < 0.05). NS: Not Significant.

**Table 5 plants-14-00609-t005:** Macronutrients of iceberg lettuce leaves grown with different biostimulants in the first trial (%).

Treatment	N	P	K	Ca	Mg
Control	2.30 b	0.26 b	2.71 c	1.94 ab	0.79 c
FA 40	2.66 ab	0.20 c	3.34 abc	2.29 a	1.34 b
FA 80	2.57 ab	0.23 bc	4.27 a	2.23 a	1.60 b
AA 75	2.36 b	0.23 bc	3.86 ab	1.68 b	1.93 a
AA 100	2.37 b	0.24 bc	3.00 bc	1.67 b	0.40 d
VC 1 mL	2.83 ab	0.25 bc	3.53 abc	1.50 bc	0.34 d
VC 2 mL	3.16 a	0.31 a	3.94 ab	1.10 c	0.53 cd
*p*	0.0039	<0.0001	<0.0001	<0.0001	<0.0001
${Tukey}_{0.05}$	0.653	0.0524	0.976	0.514	0.304

FA = Fulvic acid, AA = Aminoacid, VC = Vermicompost. No significant difference was observed between means sharing the same letter within the same column (*p* < 0.05).

**Table 6 plants-14-00609-t006:** Macronutrients of iceberg lettuce leaves grown with different biostimulants in the second trial (%).

Treatment	N	P	K	Ca	Mg
Control	2.67 bc	0.42 ab	3.61	1.03 ab	0.79 a
FA 40 + AA 100	3.03 ab	0.43 a	3.70	0.97 ab	0.70 ab
FA 40 + VC 2 mL	3.16 a	0.44 a	3.89	1.14 a	0.67 ab
AA 100 + VC 2 mL	3.00 a ab	0.40 ab	3.88	0.96 ab	0.73 a
FA40 + AA 100 + VC 2 mL	2.50 b c	0.36 b	3.44	0.64 b	0.50 b
*p*	0.0011	0.0087	0.5855	0.0321	0.0063
${Tukey}_{0.05}$	0.404	0.0654	NS	0.438	0.218

FA = Fulvic acid, AA = Aminoacid, VC = Vermicompost, NS: Non-Significant. No significant difference was observed between means sharing the same letter within the same column (*p* < 0.05). NS: Not Significant.

**Table 7 plants-14-00609-t007:** Effects of biostimulants on antioxidant of iceberg lettuce in first trial aquaculture.

Treatments	Vitamin C (mg 100 g FW^−1^)	Total Phenols (mg GA 100 g FW^−1^)	Total Flavonoids (mg RU 100 g FW^−1^)
Control	5.40 d	41.39 ab	188 b
FA 40	6.37 c	51.88 a	302 a
FA 80	6.57 bc	39.08 b	182 b
AA 75	6.76 abc	41.19 ab	218 b
AA 100	6.78 abc	45.10 ab	247 ab
VC 1 mL	7.23 a	32.94 b	178 b
VC 2 mL	6.92 ab	44.51 ab	243 ab
*p*	<0.0001	0.0055	0.0008
${Tukey}_{0.05}$	0.493	12.188	74.574

FA = Fulvic acid, AA = Aminoacid, VC = Vermicompost. No significant difference was observed between means sharing the same letter within the same column (*p* < 0.05). FW: Fresh weight, GA: Gallic acid, RU: Rutin.

**Table 8 plants-14-00609-t008:** Effects of biostimulants on antioxidant of iceberg lettuce in second trial of aquaculture.

Treatments	Vitamin C (mg 100 g FW^−1^)	Total Phenols (mg GA 100 g FW^−1^)	Total Flavonoids (mg RU 100 g FW^−1^)
Control	5.71 d	57.71 c	231 c
FA 40 + AA 100	6.57 ab	81.08 ab	338 ab
FA 40 + VC 2 mL	6.23 bc	70.57 bc	313 b
AA 100 + VC 2 mL	5.95 cd	82.56 ab	329 ab
FA 40 + AA 100 + VC 2 mL	6.69 a a	88.03 a	352 a a
*p*	0.0003	0.0013	<0.0001
${Tukey}_{0.05}$	0.447	16.142	35.437

FA = Fulvic acid, AA = Aminoacid, VC = Vermicompost. No significant difference was observed between means sharing the same letter within the same column (*p* < 0.05). FW: Fresh weight, GA: Gallic acid, RU: Rutin.

**Table 9 plants-14-00609-t009:** Effects of biostimulants on some quality parameters of iceberg lettuce in the first trial.

Treatments	Brix (%)	pH	EC (µS cm^−1^)
Control	1.77	6.64 abc	780 b
FA 40	1.55	6.66 ab	1150 a
FA 80	1.60	6.69 a	1182 a
AA 75	1.30	6.73 a	1081 a
AA 100	1.40	6.39 bc	1123 a
VC 1 mL	1.42	6.37 c	966 ab
VC 2 mL	1.90	6.68 a	1123 a
*p*	0.4389	0.0013	0.0008
${Tukey}_{0.05}$	NS	0.286	260.679

FA = Fulvic acid, AA = Aminoacid, VC = Vermicompost. No significant difference was observed between means sharing the same letter within the same column (*p* < 0.05). NS: Not Significant.

**Table 10 plants-14-00609-t010:** Effects of biostimulants on some quality parameters of iceberg lettuce in the second trial.

Treatments	Brix (%)	pH	EC (µS cm^−1^)
Control	2.25	6.56 ab	1344 b
FA 40 + AA 100	2.50	6.43 b	1697 ab
FA 40 + VC 2 mL	2.30	6.62 a	1569 ab
AA 100 + VC 2 mL	2.50	6.46 ab	1952 a
FA 40 + AA 100 + VC 2 mL	2.54	6.60 ab	1618 ab
*p*	0.4382	0.0263	0.0440
${Tukey}_{0.05}$	NS	0.188	537.117

FA = Fulvic Acid, AA = Aminoacid, VC = Vermicompost. No significant difference was observed between means sharing the same letter within the same column (*p* < 0.05). NS: Not Significant.

## Data Availability

The data supporting the findings of this study are included within the article.

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
