# Peer review of "Fulvic Acid, Amino Acids, and Vermicompost Enhanced Yield and Improved Nutrient Profile of Soilless Iceberg Lettuce"

_plants, 2025, doi:10.3390/plants14040609_

Round 1
Reviewer 1 Report
Comments and Suggestions for Authors
The manuscript entitled ‘Fulvic Acid, Amino Acids, and Vermicompost Improved Enhanced Yield, and Nutrient Profile of Soilless Iceberg Lettuce’ fits within the general scope of Plants MDPI. The research paper explores the effects of fulvic acid, amino acid, and vermicompost biostimulants on the growth, yield, and nutrient profile of soilless grown iceberg lettuce (Lactuca sativa var. capitata) in floating culture under controlled glasshouse conditions.
The authors found that applying eco-friendly biostimulants, such as fulvic acids, amino acids and vermicompost, both individually and in combination, improved key plant parameters, including biomass production, nutrient uptake, and antioxidant content, while reducing nitrate levels in lettuce leaves.
The manuscript is original and it is of high significance to the field and of high interest to the general audience. However, there are some major concerns. The authors declared that two trials were carried oud in two DIFFERENT PERIODS. Since plants differently react to biostimulants depending on growing conditions and considering that the second trial was conducted in a different season, how could the first trial represent a model to assess the optimal biostimolant dose? If authors want to offer a complete overview of the effect of biostimulants on lettuce plants cultivated under greenhouse, I would like to advise including all biostimulants, alone or combined, in both growing cycles, pointing out via statistical analysis, any different behavior observed in the two trials (please include the growing season as fix factor).
Scientific name in Italics.
Reviewer 2 Report
Comments and Suggestions for Authors
This paper investigates the effects of different products and combinations on soilless cultivated plants. This work is of great significance for further sustainable agriculture.
However, there are many details that need to be disclosed, such as the organic matter content of Vermicompost products, including the content of fulvic acid and humic acid, as well as the source of raw materials. Because these data will affect the actual results of the experiment.
Additionally, some bar charts are not necessary. For example, the data in figures 1 and 2 can be arranged in a tabular format, which makes the results more compact and easier to compare between processes.
Reviewer 3 Report
Comments and Suggestions for Authors
Review of the Manustript plants-3388357
In this manuscript the authors present the results of a study that investigated the effects of fulvic acid, vermicompost and amino acid biostimulants on the growth, yield and quality of iceberg lettuce grown in a hydroponic floating culture system. The topic is very interesting and provides important data on the suggestions that biostimulants, used individually or in combination, can have a synergistic effect on plant traits.
The introduction is well written, but the following sections have the following shortcomings
M&M:
What was the lights conditions during the growing periods? Please add the informations about the light conditions during the growing periods!
L559-560: How the authors explaine the increase of the EC values during the growing period? How often was the nutrient solution changed during the 74 and 70 days of the growth period in the first and second trials?
L594-597: Please check the description of the analysis of nitrate concentration in lettuce leaves. The reference you cited (28) Dasgan et al. (2023) did not use the same analytical method for nitrate determination in leaves of bell pepper plants!
L626-629: The statistical method (LSD test) was not suitable to determine the statistical differences between treatments when the number of treatments is higher than here (3). In this study, 7 (in the first trial) and 5 (in the second trial) treatments were administered. Please correct! Use the appropriate test (Tukey test) to determine the significant differences!
Results:
General suggestion regarding the Figures: Figures are presented without the title of y axis. Please correct at all Figures.
L 116-147:
Ad 2.1. Please comment only on the statistical differences and correct Figure 1 and Figure 2 where the y-axis heading is missing. Please correct the type of diagram in which the average values of the individual treatments are shown from the hystogram to the point (marker) with SE bars.
L116-129; 134-147. Please correct the comments of the Iceberg lettuce yield according to the new statistical analysis, using Tukey (HSD) test
Please describe in M&M how the iceberg lettuce yield was calculated. According to the results of the average plant weight, the cultivation area (1.05m x0.55m), the plant density (10 plants/treatment), the results for the Yiled (kg/m2) are too high! Please check and correct!
L 473-474:... have proves effective in enhancing BRIX,....this is not true (See Table 12: Brix . NS). Please correct!
L493-494the high temperature at the end of the second experiment is not the only reason for the lower nitrate content in the lettuce plants of the second experiment. The main reason was also the different light conditions, which are very important when presenting the results of the nitrate content in leafy vegetables. Please add some information (day length, intensity of solar radiation, ...) about the light conditions during the growth periods of the trials.
L. 495: typo (Alexander et al....
Round 2
Reviewer 3 Report
Comments and Suggestions for Authors
Dear Authors
I am satisfied with the corrections you have made to the article, taking into account my suggestions to improve the quality of the article.
I therefore suggest that the editor accept your MS for publication in the journal Plant